# MODELNET40-E: AN UNCERTAINTY-AWARE BENCHMARK FOR POINT CLOUD CLASSIFICATION

## ABSTRACT

Most existing benchmarks for point cloud classification focus solely on accuracy, overlooking critical aspects such as calibration and uncertainty awareness that are essential for safety-critical applications. We introduce ModelNet40-E, a benchmark that complements prior robustness efforts by providing noise-corrupted point clouds together with per-point uncertainty annotations via Gaussian parameters $(\sigma, \mu)$. Unlike benchmarks based on random corruptions, ModelNet40-E introduces physically motivated LiDAR-like noise at multiple levels, reflecting real-world sensing conditions. Using this benchmark, we evaluate a range of representative point cloud architectures across varying noise levels. Our results show that accuracy alone can be misleading: some models with lower accuracy exhibit better calibration and uncertainty awareness, underscoring the need to evaluate all three metrics together.

## 1 INTRODUCTION

In recent years, 3D point cloud classification has gained increasing importance across a wide range of applications, including autonomous driving (Lang et al., 2018; Yan et al., 2018; Sun et al., 2019), robotics (Mousavian et al., 2019), augmented reality (Izadi et al., 2011; Newcombe et al., 2015; Jiang et al., 2025; Newcombe et al., 2011; Yu et al., 2020), and remote sensing (Wang et al., 2025; 2024; Pan et al., 2025). In response, numerous models have been proposed, ranging from early architectures like PointNet (Qi et al., 2017a) and PointNet++ (Qi et al., 2017b) to more advanced designs such as DGCNN (Wang et al., 2019), Point2Vec (Abou Zeid et al., 2023), and Point Transformer (Zhao et al., 2021; Wu et al., 2022; 2024).

Constrained by the limited availability of real-world annotated point cloud data, most of these models are trained and evaluated on synthetic datasets, such as ModelNet40 (Wu et al., 2015) and Shapenet (Chang et al., 2015). These datasets are composed of clean, idealized point clouds that lack the noise, occlusions, and distortions typically present in real sensor data. As a result, models that perform well in these benchmarks often fail to generalize to real-world environments (Wu et al., 2018; Zhao et al., 2022; Ren et al., 2025).

To ensure fair comparisons and improve the applicability of the models in real-world scenarios, robustness benchmarks like ModelNet-C (Sun et al., 2022) have been proposed. These benchmarks introduce synthetic corruptions such as rotations, jittering, scaling, and point dropout to test the resilience of models to input perturbations. Despite offering valuable insights, they face key limitations: (1) they apply generic perturbations that do not reflect realistic LiDAR noise, and (2) they lack ground-truth uncertainty annotations. In safety-critical applications, such as autonomous driving and robotic manipulation, it is not enough for a model to simply output a class label. Equally important is assessing how confident the model is in its predictions. Overconfident yet incorrect predictions can lead to catastrophic failures, making calibration and uncertainty awareness essential attributes for building reliable models.

We introduce ModelNet40-E, a novel benchmark that builds upon the original ModelNet40 dataset by corrupting it with LiDAR-like noise at three severity levels: Light, Moderate, and Heavy. Unlike random jitter or dropout, LiDAR noise is range-dependent, anisotropic, and often structured, influenced by surface reflectivity, incidence angles, and environmental conditions. Capturing these complexities is critical for creating realistic benchmarks that truly test model robustness and uncertainty estimation. Our noise simulation is inspired by empirical studies of LiDAR measurement

errors (Gschwandtner et al., 2011), incorporating range-dependent Gaussian noise, incidence angle effects, systematic bias, and random outliers. Crucially, for every point in the dataset, we also provide the corresponding per-point noise statistics: standard deviation $\sigma$ and expected bias $\mu$. This allows for rigorous evaluation of both predictive performance and uncertainty estimation.

The main contributions of this work are as follows:

- We introduce **ModelNet40-E**, a benchmark for point cloud classification that extends the standard ModelNet40 dataset with realistic LiDAR-inspired noise, enabling evaluation not only of classification accuracy but also of *calibration*, and *uncertainty awareness*.
- We propose a **LiDAR-like noise generation procedure** that injects range-dependent, angle-dependent, and outlier noise, providing controlled yet realistic perturbations for benchmarking robustness.
- Using ModelNet40-E, we conduct a systematic evaluation of several representative architectures, highlighting that while performance degrades across noise levels, models differ significantly in robustness, calibration, and error detection ability.

## 2 RELATED WORK

### 2.1 POINT CLOUD CLASSIFICATION BENCHMARKS

3D point cloud classification models are commonly evaluated on synthetic datasets such as ModelNet40 (Wu et al., 2015) and Shapenet (Chang et al., 2015). Our work focuses on ModelNet40, a widely used benchmark consisting of 12,311 CAD-generated point clouds across 40 categories. The dataset is split into training ($80\%$) and testing ($20\%$) sets, containing 9,840 and 2,468 samples, respectively. It contains clear and uniformly sampled point clouds, making it easily processed and suitable as a standard benchmark.

However, such synthetic datasets do not accurately reflect real-world sensor data, which is often affected by noise, occlusions, and incomplete scans. Consequently, models trained solely on synthetic data often struggle to generalize to real-world scenarios.

To address this limitation, Sun et al. (2022) proposed ModelNet40-C, a corrupted version of ModelNet40 that introduces a wide range of synthetic perturbations, including Gaussian noise, scaling, rotation, translation, point dropout, and jitter. These corruptions are designed to simulate various distortions that models might encounter in real-world data. Sun et al. (2022) evaluate robustness by measuring classification accuracy under each corruption type and severity level.

### 2.2 ROBUSTNESS AND ADVERSARIAL ATTACKS ON POINT CLOUDS

Robustness in point cloud classification refers to a model's ability to maintain performance under various perturbations. These include sensor noise, occlusions, point sparsity, and misalignment, which are common challenges present in real-world data. To evaluate resilience under such conditions, recent benchmarks like ModelNet40-C (Sun et al., 2022) apply a wide range of synthetic corruptions (e.g., rotation, scaling, dropout, jitter) and measure performance across different perturbation levels.

Beyond natural corruptions, there has been growing interest in studying adversarial robustness, where small, carefully crafted perturbations are designed to fool classifiers. Several works have extended adversarial attack strategies from the 2D image domain to 3D point clouds. For instance, Xiang et al. (2018) proposes adversarial perturbation and point generation strategies to mislead PointNet into high-confidence misclassifications. In addition, Zheng et al. (2018) introduces a point perturbation attack that uses gradient information to identify and remove critical points, degrading classification accuracy. Other approaches, like Liu et al. (2019) and Tsai et al. (2020), have explored transformations and translations to mislead classifiers while preserving overall object shape.

To counter these attacks, a variety of defense strategies have been developed, including adversarial training, which improves model robustness by exposing it to adversarial examples during training (Sun et al., 2020), as well as input reconstruction methods that denoise or repair corrupted point clouds before classification (Zhou et al., 2018). However, many of these defenses come at the cost

of reduced clean accuracy, and few have been evaluated under realistic sensor noise conditions, limiting their practical effectiveness.

While adversarial attacks expose important model vulnerabilities, robustness to natural corruptions is arguably more critical for real-world deployment. In practical applications such as autonomous driving, models must remain reliable in the presence of LiDAR noise, occlusions, and imperfect sensor inputs. However, most existing benchmarks focus on synthetic noise, often failing to capture the complexity of real-world 3D sensing environments.

### 2.3 UNCERTAINTY ESTIMATION IN DEEP LEARNING

Uncertainty estimation in deep learning refers to a model's ability to not only make predictions but also to quantify how confident it is in those predictions. In a well-calibrated model, confidence scores should decrease appropriately as input perturbations are introduced. If confidence scores remain high even in the presence of significant corruptions, this may lead to overconfident mis-classifications. It is well established that neural networks are often miscalibrated and tend to be overconfident, particularly as model capacity or training time increases (Guo et al., 2017).

Uncertainty is often categorized into two types: *aleatoric* and *epistemic*. Aleatoric uncertainty refers to the inherent noise in the data, such as sensor measurement errors or occlusions in point clouds. This type of uncertainty cannot be reduced by introducing more data. In contrast, epistemic uncertainty arises from the model's ignorance due to limited or imbalanced data and can be reduced by improving data diversity or quantity.

Several techniques exist to estimate these uncertainties. For example, Gal & Ghahramani (2016) proposes the Monte Carlo (MC) dropout method, which performs multiple stochastic forward passes with dropout enabled at test time to approximate predictive uncertainty. In addition, deep ensembles (Lakshminarayanan et al., 2017) train multiple models with different initial conditions and estimate uncertainty by measuring the variance between their outputs.

## 3 MODELNET40-E

### 3.1 BENCHMARK OVERVIEW

ModelNet40-E is an uncertainty-aware benchmark for point cloud classification. It is built upon the original ModelNet40 dataset (Wu et al., 2015), introducing several levels of LiDAR-like noise based on realistic sensor noise. By providing per-point noise statistics ($\sigma$ and $\mu$), the main goal of ModelNet40-E is to serve as a common benchmark for evaluating the uncertainty estimation capabilities of point cloud classification models.

We introduce three different levels of noise to the original ModelNet40 dataset: Light, Moderate, and Heavy. The number of samples and categories are identical to the original dataset, as well as the training and test splits.

### 3.2 LIDAR NOISE SIMULATION

To approximate realistic sensor noise, we simulate a LiDAR acquisition process inspired by empirical studies of LiDAR measurement errors (Gschwandtner et al., 2011). For each clean point cloud, we compute per-point range and incidence angle relative to the sensor. Measurement noise is sampled from a Gaussian distribution whose standard deviation increases with distance and varies systematically with surface orientation. Specifically, the per-point noise $\epsilon$ is sampled as

$$\epsilon \sim \mathcal{N}\big(\mu(\theta), \sigma(r, \theta)\big),$$

where $r$ denotes the distance to the sensor, and $\theta$ is the incidence angle between the incoming ray and the estimated local surface normal. In particular, the noise model includes the following parameters:

**Range-dependent noise.** LiDAR noise increases with range. To model this, we define the range-dependent noise as

$$\sigma_{range} = a + b \cdot r,$$

where $r$ is the distance from the sensor to the point, $a$ is a constant base noise term (in meters), and $b$ scales noise linearly with distance.

Table 1: Configuration parameters for the light, moderate, and heavy noise levels in ModelNet40-E. For each point cloud, the parameters $(a, b)$, $c$, $k$, and the outlier probability are sampled uniformly from the ranges shown in the table.

| Noise level | Range Noise $(a, b) \times 10^3$ | Angle Factor $c$ | Bias $k \times 10^3$ | Outlier prob. (%) |
|---|---|---|---|---|
| Light | ([2.0, 4.0], [0.5, 1.5]) | [1.0, 2.0] | [2.5, 7.5] | [0.5, 1.5] |
| Moderate | ([3.0, 7.0], [1.0, 3.0]) | [1.5, 2.5] | [5.0, 15.0] | [1.0, 3.0] |
| Heavy | ([5.0, 15.0], [2.0, 4.0]) | [2.0, 4.0] | [10.0, 25.0] | [4.0, 8.0] |

**Angle-dependent noise.** Noise in LiDAR data also depends on the angle at which the laser hits the surface. The more obliquely the angle, the greater the noise. We model this effect as

$$\sigma_{angle} = 1 + c \cdot (1 - \cos\theta),$$

where $\theta$ is the incidence angle between the ray and the surface normal, and $c$ is the scaling factor. Higher $c$ means greater sensitivity to the incidence angle.

**Bias ($k$).** In addition to range- and angle-dependent measurement noise, we include a systematic bias term related to sensor calibration. This bias is modeled as

$$\mu = k \cdot (1 - \cos\theta),$$

where $\theta$ is the incidence angle and $k$ controls the magnitude of the offset.

**Outlier probability.** Outliers are common in LiDAR data, often caused by spurious reflections, multipath returns, or transient measurement artifacts. We define an outlier probability parameter specifying the proportion of points that are replaced with random points sampled uniformly within the normalized bounding box $[-0.5, 0.5] \times [-0.5, 0.5] \times [-0.5, 0.5]$ around the object.

This procedure yields perturbed point clouds along with the ground-truth noise statistics (mean bias $\mu$ and standard deviation $\sigma$) per point.

### 3.3 Noise Configurations

We consider four noise configurations: None, Light, Moderate, and Heavy. The *None* setting corresponds to the original, unperturbed ModelNet40 dataset. For the corrupted versions, the parameters are sampled from the ranges summarized in Table 1. During noise simulation, the virtual sensor position is randomly sampled on a sphere of radius 2.0, with azimuth uniformly drawn from $[0, 2\pi]$ and elevation from $[-\pi/4, \pi/4]$.

## 4 Experiments

### 4.1 Experimental Setup

We evaluate our benchmark on several representative architectures for point clouds: PointNet (Qi et al., 2017a), PointNet++ (Qi et al., 2017b), DGCNN (Wang et al., 2019), SimpleView (Goyal, 2020), CurveNet (Xiang et al., 2021), PointMLP (Ma et al., 2022), and Point Transformer v3 (PTv3; Wu et al. 2024). Each model is trained independently on the original ModelNet40 training dataset ("None" noise level in our paper), and evaluated on each noise level ("None", "Light", "Moderate", and "Heavy") test set. We report classification accuracy, Expected Calibration Error (ECE), AUROC (Error Detection), and Pearson correlation between ground-truth measured and predicted uncertainty.

All models are trained for 100 epochs using cross-entropy loss and the Adam optimizer (Kingma, 2014), with an initial learning rate of $1 \times 10^{-3}$ decayed by a factor of 0.7 every 20 epochs. The batch size is 32 for all models except PTv3, which uses a reduced batch size of 8 due to its higher memory requirements. To ensure fair comparison across accuracy, calibration, and uncertainty awareness, we consistently evaluate the checkpoint from the final epoch (epoch 100) rather than selecting models based on accuracy alone. All experiments were conducted on a single NVIDIA RTX 4090 GPU.

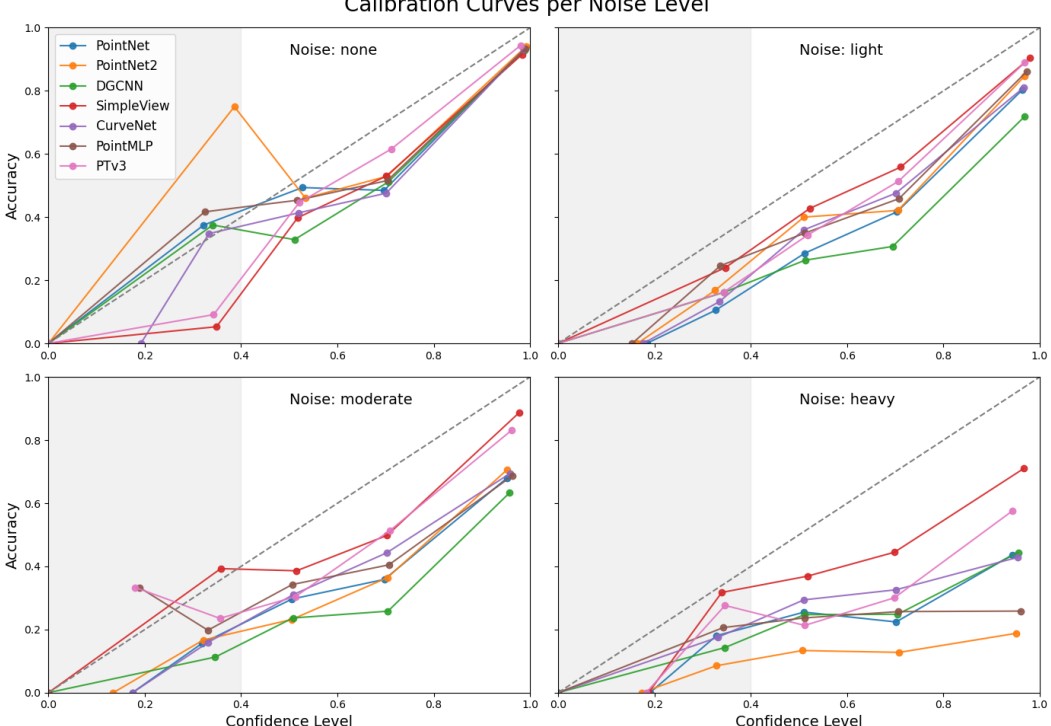

Figure 1: Calibration curves for all models across different noise levels. The dashed diagonal line represents perfect calibration. The gray shaded area indicates confidence values under $40\%$, which are underrepresented and less reliable.

## 4.2 EVALUATION METRICS

### 4.2.1 CLASSIFICATION ACCURACY

Accuracy evaluates the predictive performance of classification models. It is defined as the proportion of correctly predicted test samples. In our experiments, we report accuracy in the test set for each noise level independently. It is expected that accuracy decreases as the noise level increases, and a lower decrement signals better robustness to noise.

### 4.2.2 EXPECTED CALIBRATION ERROR (ECE)

ECE (Hackel et al., 2017; Naeini et al., 2015; Nixon et al., 2019) quantifies the discrepancy between a model's predicted confidence and its actual accuracy. In a perfectly calibrated model, the predicted confidence matches the observed accuracy. For instance, predictions made with $70\%$ confidence should be correct approximately $70\%$ of the time.

In practice, we compute ECE by dividing the predictions into $M$ bins and comparing the average confidence and accuracy within each bin:

$$\text{ECE} = \sum_{i=1}^{M} \frac{n_i}{N} \left| \text{acc}_i - \text{conf}_i \right|,$$

where $n_i$ is the number of predictions in the $i^{\text{th}}$ bin, and $N$ is the total number of predictions. $\text{acc}_i$ and $\text{conf}_i$ represent the average accuracy and confidence in the $i^{\text{th}}$ bin, respectively.

### 4.2.3 UNCERTAINTY AWARENESS

We evaluate uncertainty awareness using two complementary measures.

Table 2: Classification Accuracy (%) for all models across different noise levels. Highest accuracy under each noise severity is highlighted in bold.

| Model | None | Light | Moderate | Heavy |
|---|---|---|---|---|
| PointNet | 88.86 | 65.11 | 53.69 | 33.79 |
| PointNet++ | **91.61** | 70.91 | 52.84 | 16.21 |
| DGCNN | 88.78 | 60.09 | 50.49 | 36.99 |
| SimpleView | 85.62 | **84.60** | **80.47** | **63.01** |
| CurveNet | 89.14 | 69.12 | 58.55 | 38.09 |
| PointMLP | 89.99 | 76.01 | 59.08 | 25.32 |
| PTv3 | 89.47 | 79.42 | 71.11 | 43.84 |

(1) *AUROC.* AUROC is a standard metric that evaluates the model's ability to detect errors (incorrectly classified samples) using its predicted uncertainty. Specifically, we define the predicted uncertainty as $1 - p_{\max}$, where $p_{\max}$ is the maximum softmax probability assigned to any class. The ROC curve is constructed by plotting the true positive rate against the false positive rate at varying uncertainty thresholds, where incorrect predictions are treated as the positive class (label 1) and correct predictions as the negative class (label 0). The AUROC corresponds to the area under this curve. A higher AUROC indicates that the model assigns larger uncertainty values to its errors than to its correct predictions, reflecting stronger error-detection capability.

(2) *Pearson Correlation Between Ground-Truth Measurement Uncertainty and Predicted Uncertainty.* A central contribution of this work is introducing a new measure of *uncertainty awareness*, made possible by the ground-truth noise statistics provided in ModelNet40-E. Each point cloud sample comes with a measurement uncertainty $\sigma$ that reflects the standard deviation of the LiDAR-like noise applied during data generation. This allows us to directly test whether a model's predicted uncertainty increases when the true measurement uncertainty increases.

To quantify this relationship, we compute the Pearson correlation coefficient between ground-truth measurement uncertainty $X$ and predicted uncertainty $Y$:

$$r = \frac{\mathrm{cov}(X, Y)}{\sigma_X \, \sigma_Y}.$$

Predicted uncertainty is defined as $1 - p_{\max}$, where $p_{\max}$ is the maximum softmax probability assigned to any class. A high correlation indicates that the model assigns higher uncertainty to noisier inputs, demonstrating strong uncertainty awareness.

However, measuring correlation on all samples mixes two effects: genuine uncertainty awareness and the natural confidence drop under noise. To isolate the true awareness signal, we compute the Pearson correlation using only correctly classified samples, capturing whether the model's confidence decreases appropriately as noise increases while still predicting the correct class. For completeness, we also report the correlation over all samples for comparison.

### 4.3 MAIN RESULTS

#### 4.3.1 CLASSIFICATION ACCURACY

Table 2 reports classification accuracy across all models and noise levels. As expected, performance degrades as noise severity increases. Early architectures such as PointNet, PointNet++ and DGCNN experience sharp degradation under noise, whereas more recent methods such as SimpleView and PTv3 maintain a higher accuracy under noise, indicating stronger robustness to perturbations. Notably, SimpleView achieves the lowest accuracy on the clean dataset but the highest accuracy under light, moderate, and heavy noise, highlighting its resilience to corruption.

#### 4.3.2 EXPECTED CALIBRATION ERROR (ECE)

Table 3 reports the Expected Calibration Error across all models and noise levels. As with accuracy, calibration deteriorates as noise severity increases, indicating that higher measurement uncertainty not only reduces predictive performance but also leads to less reliable confidence estimates.

Table 3: Expected calibration error for all models across noise levels. Lower is better.

| Model | None | Light | Moderate | Heavy |
|---|---|---|---|---|
| PointNet | 0.0628 | 0.1897 | 0.2671 | 0.4269 |
| PointNet++ | 0.0564 | 0.1441 | 0.2629 | 0.6417 |
| DGCNN | 0.0708 | 0.2664 | 0.3325 | 0.4550 |
| SimpleView | 0.0800 | **0.0828** | **0.1048** | **0.2392** |
| CurveNet | 0.0635 | 0.1666 | 0.2512 | 0.4400 |
| PointMLP | 0.0626 | 0.1294 | 0.2613 | 0.5835 |
| PTv3 | **0.0437** | 0.1004 | 0.1488 | 0.3492 |

Table 4: Uncertainty awareness results across all models. We report AUROC for error detection under different noise levels (None, Light, Moderate, Heavy), and Pearson correlation between ground-truth measurement uncertainty $\sigma$ and predicted uncertainty $(1 - p_{\max})$. Correlations are calculated after pooling samples across all noise levels, including only correctly classified samples (correct only), and all samples (all). Higher AUROC and Pearson values indicate stronger error detection and uncertainty awareness, respectively.

| Model | AUROC (None) | AUROC (Light) | AUROC (Moderate) | AUROC (Heavy) | Pearson (correct only) | Pearson (all) |
|---|---|---|---|---|---|---|
| PointNet | 0.8971 | 0.8367 | 0.7650 | 0.6584 | 0.1787 | 0.1180 |
| PointNet++ | **0.9081** | 0.8446 | 0.7949 | 0.5825 | 0.1400 | 0.0463 |
| DGCNN | 0.9043 | 0.8102 | 0.7782 | 0.6574 | 0.1334 | 0.0754 |
| SimpleView | 0.8746 | 0.8498 | **0.8306** | 0.7230 | 0.1291 | 0.1397 |
| CurveNet | 0.8852 | 0.8138 | 0.7353 | 0.5885 | 0.1295 | 0.0520 |
| PointMLP | 0.8798 | 0.8526 | 0.7309 | 0.5178 | **0.2003** | 0.0918 |
| PTv3 | 0.8980 | **0.8653** | 0.8282 | **0.7292** | 0.1853 | **0.2118** |

PTv3 achieves the best calibration on the clean set, even if it did not achieve the highest accuracy. Under noise, both PTv3 and SimpleView present the lowest ECE, with SimpleView performing slightly better. Notably, while both PointNet++ and PointMLP maintain a relatively low ECE under clean and light noise levels, suffer from a sharp degradation under moderate and heavy noise levels, a behavior that is consistent with their accuracy results.

Figure 1 shows the calibration curves for all models. The dashed diagonal indicates perfect calibration, while the gray shaded area indicates low-confidence predictions ($< 40\%$), which are underrepresented and thus less reliable. Across all noise configurations, models tend to become increasingly overconfident as noise severity rises, with curves falling below the diagonal. This effect is particularly notable under moderate and heavy noise.

### 4.3.3 UNCERTAINTY AWARENESS

Table 4 reports the results for uncertainty awareness, including (i) AUROC and (ii) Pearson correlation between ground-truth measurement uncertainty $\sigma$ and predicted uncertainty.

(1) *AUROC*. It measures the ability of each model to distinguish between correct and incorrect predictions using predicted uncertainty. Across all architectures, AUROC decreases as noise severity increases, indicating that error detection becomes more difficult under stronger perturbations. Among the models, PointNet++ achieves the highest AUROC on clean data, while SimpleView achieves the best performance under moderate noise. PTv3 is the most consistent overall, ranking near the top across all noise levels and achieving the highest AUROC under light and heavy noise.

(2) *Correlation Between Measurement Uncertainty and Predicted Uncertainty*. This metric evaluates whether models reduce their confidence as measurement noise increases. To isolate genuine uncertainty awareness from the trivial confidence drop caused by misclassifications, we report correlations on correctly classified samples as our primary metric (see Section 4.2). For completeness, we also include results on all samples, providing additional insights into whether models remain uncertainty-aware even when making incorrect predictions.

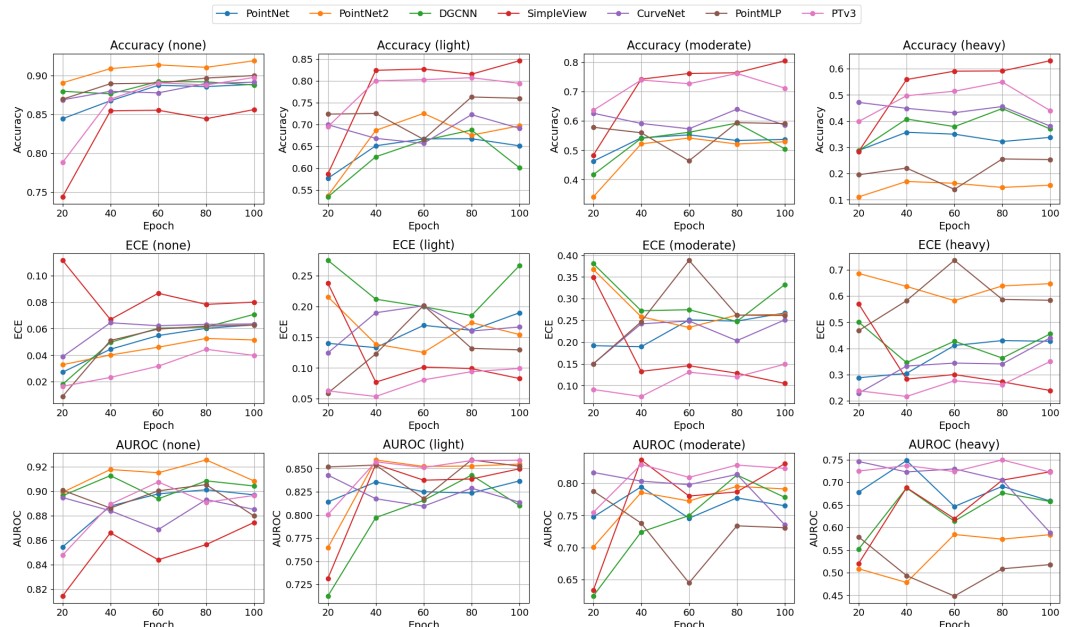

Figure 2: Training dynamics of accuracy (top row), Expected Calibration Error (ECE, middle row), and AUROC (bottom row) for all models across training epochs under varying noise levels.

PointMLP achieves the strongest correlation on correctly classified samples, followed by PTv3 and PointNet, indicating that these models adjust their confidence most consistently in response to increasing noise while still making correct predictions. However, PointMLP's correlation drops substantially when all samples are included, suggesting that while it is highly sensitive to noise on correct predictions, its uncertainty estimates do not reliably capture misclassified inputs. In contrast, PTv3 achieves both strong correct-only correlation and the highest all-sample correlation, demonstrating more consistent uncertainty-awareness, which is also supported by its superior AUROC results.

SimpleView, despite achieving excellent robustness under noise for accuracy and ECE, shows a relatively weak correlation, implying that its confidence estimates are less sensitive to input noise.

### 4.3.4 TRAINING DYNAMICS

We investigate the training dynamics for all models by evaluating accuracy, Expected Calibration Error (ECE), AUROC, and Pearson correlation between ground-truth and predicted uncertainty across several training times and noise levels. Each model is assessed after 20, 40, 60, 80, and 100 training epochs.

Figure 2 shows our results for accuracy, ECE, and AUROC, presented by noise level. As training progresses, accuracy generally improves across models, but this improvement often comes at the cost of higher calibration error, particularly under moderate and heavy noise. This effect is particularly evident for SimpleView. AUROC trends are more stable, though some models such as PointMLP show notable fluctuations under moderate and heavy noise levels.

Figure 3 presents the dynamics of the Pearson correlations, for both correct-only and all samples. In this case, the results are derived after pooling samples across all noise levels. On correctly classified samples, most models initially achieve moderate correlation values that gradually decline with training, suggesting a loss of sensitivity to input noise, particularly for PointNet++ and DGCNN. When including all samples, correlations are typically lower, reflecting the impact of misclassifications. Interestingly, PTv3 maintains relatively strong and stable correlations in both settings, while other models such as PointMLP show strong correct-only correlations but collapse when misclassified samples are included.

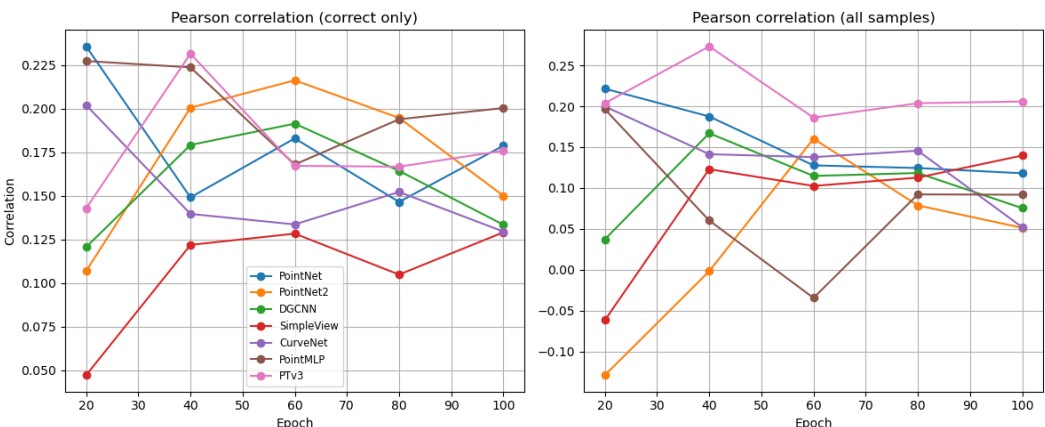

Figure 3: Training dynamics of Pearson correlation between ground-truth measurement uncertainty and predicted uncertainty across training epochs, computed on correctly classified samples (left) and on all samples (right). Correlations are calculated after pooling samples across all noise levels.

These results highlight a key trade-off: longer training improves accuracy but can degrade both calibration and uncertainty awareness. Such a trend aligns with prior observations in the literature (Guo et al., 2017). Monitoring metrics beyond accuracy—particularly ECE and correlation-based uncertainty measures—provides a more complete picture of model reliability under noise.

## 5 CONCLUSION

In this work, we introduced ModelNet40-E, an uncertainty-aware benchmark for point cloud classification. Unlike existing benchmarks that evaluate models solely on accuracy, ModelNet40-E enables systematic assessment of calibration and uncertainty awareness under LiDAR-like noise. By providing ground-truth measurement uncertainty alongside corrupted point clouds, ModelNet40-E enables a clear separation between robustness, calibration, and uncertainty awareness.

Our evaluation across multiple representative architectures highlights several key findings. First, robustness to noise varies significantly: while early models such as PointNet and DGCNN suffer sharp degradation, more recent approaches like SimpleView and PTv3 maintain higher accuracy under corruption. Second, calibration errors grow substantially with noise, but transformer-based PTv3 demonstrates consistently superior calibration compared to other models. Finally, our new correlation-based metric reveals notable differences in uncertainty awareness. PTv3 achieves both strong correct-only and all-sample correlations, reflecting reliable noise sensitivity, whereas models like PointMLP exhibit strong awareness on correct predictions but fail to capture uncertainty on misclassified samples.

## LLM USAGE DISCLOSURE

This paper benefited from the use of large language models (LLMs) to assist with writing, code development, and literature research. All outputs generated by LLMs were carefully reviewed and verified by the authors, and all scientific contributions were developed by the authors.

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
