# OpenReview forum: "ModelNet40-E: An Uncertainty-Aware Benchmark for Point Cloud Classification"
_ICLR.cc/2026/Conference — Submitted to ICLR 2026_

### Official Review · Reviewer_k8f4 · 2025-10-25

**Soundness:** 2
**Presentation:** 2
**Contribution:** 2
**Rating:** 2
**Confidence:** 3

**Summary:**

The paper proposed ModelNet40-E, a new 3D point cloud classification benchmark which provides two new features: (1) simulated realistic LiDAR noise on the point clouds, and (2) provided point-wise ground truth uncertainty for evaluating uncertainty quantification tasks.

**Strengths:**

+ The uncertaity assessment is important for machine learning tasks, so does the point cloud classification.

**Weaknesses:**

- The demonstration of the dataset samples is missing. While the paper claim realistic LiDAR noise models, there is no figures illustrate the benchmark dataset samples, making the reader not easy to assess how different the samples are from original perfect point clouds. Such figures are commonly provided by ModelNet40 variants in their papers.
- Though LiDAR noise is injected, the paper did not detail how the point clouds are sampled from the 3D models. I assume the "realistic" LiDAR images should be simulated by ray casting on the 3D model, with selected LiDAR configurations and locations, preserving the pattern of LiDAR sweeps. Otherwise, the data sample is still not like real LiDAR results. I did not find details in paper about it.
- While uncertainty quantification evaluation is interesting, I am not sure whether ModelNet40-E is sufficient for that purpose. There is only a few selections of type and magtitude of noise (low, moderate, and high), while the uncertainty quantification should be evaluated on a diversity of noise setups.

**Questions:**

* How are the point clouds sampled before injecting noises?

---

### Official Review · Reviewer_QtXM · 2025-10-30

**Soundness:** 2
**Presentation:** 2
**Contribution:** 2
**Rating:** 2
**Confidence:** 5

**Summary:**

This paper introduces ModelNet40-E, a benchmark designed to evaluate not only classification accuracy but also calibration and uncertainty awareness in 3D point cloud classification. Unlike prior benchmarks such as ModelNet40-C, which apply generic corruptions (e.g., rotation, jitter, dropout), ModelNet40-E injects physically motivated LiDAR-like noise at multiple severity levels (light, moderate, heavy).

The noise model incorporates range-dependent Gaussian variance, angle-dependent perturbations, systematic bias, and outlier probabilities, yielding point clouds with realistic distortions along with per-point noise statistics (µ, σ).

The benchmark is used to evaluate seven models, trained on clean ModelNet40 and tested under corrupted conditions. Metrics include accuracy, Expected Calibration Error (ECE), AUROC for error detection, and a new measure—Pearson correlation between ground-truth and predicted uncertainty. Results highlight that robustness varies widely across models.

**Strengths:**

- The focus on calibration and uncertainty awareness in addition to accuracy is important for real-world safety-critical applications, where overconfident misclassifications can be harmful. The LiDAR-inspired corruption (range-dependent noise, incidence-angle effects, bias, and outliers) goes beyond simple jittering or dropout, making this benchmark more physically grounded.

- Results reveal trade-offs between clean accuracy and robustness. For instance, SimpleView, despite weaker clean accuracy, consistently outperforms others under corruption, while PTv3 shows both strong calibration (lowest ECE) and stable uncertainty correlation.

**Weaknesses:**

- While ModelNet40-E extends a classic dataset, it is still synthetic, small-scale data. Conclusions may not fully carry over to large-scale real-world LiDAR datasets (e.g., nuScenes, Waymo, SemanticKITTI). Additionally, there are several closely related works that were not discussed: Uncertainty Estimation and Out-of-Distribution Detection for LiDAR Scene Semantic Segmentation (ECCV 2024), Calib3D: Calibrating Model Preferences for Reliable 3D Scene Understanding (WACV 2025), and MSC-Bench: Benchmarking and Analyzing Multi-Sensor Corruption for Driving Perception (ICME 2025).

- Limited architectural diversity: The benchmark covers seven models, but all are classification-focused; inclusion of self-supervised or pretraining-heavy backbones (e.g., Point-BERT, Point-MAE) would strengthen the evaluation.

- Noise realism: Although LiDAR-inspired, the corruption is still simulated; validating noise statistics against real sensor logs (e.g., nuScenes, KITTI) would improve credibility.

- The Pearson correlation between noise level and uncertainty is a useful addition, but its interpretation is not straightforward—low correlation may reflect either poor awareness or confounding effects from misclassifications.

- The benchmark requires storing and using per-point µ, σ annotations, which increases data size. Discussion of computational overhead is limited.

**Questions:**

1. How closely do your noise distributions (σ, µ ranges) match empirical LiDAR datasets such as Waymo or nuScenes?

2. Would training on noisy data (rather than just clean) change the rankings of models in robustness and calibration?

3. How sensitive are the results to the choice of hyperparameters for the noise model (e.g., base noise term a, angle factor c)?

4. Can the benchmark be extended beyond classification—for example, to segmentation or detection tasks where uncertainty is also critical?

---

### Official Review · Reviewer_guUM · 2025-10-31

**Soundness:** 2
**Presentation:** 2
**Contribution:** 2
**Rating:** 2
**Confidence:** 4

**Summary:**

This manuscript introduces ModelNet40-E, a benchmark to evaluate calibration and uncertainty awareness in point cloud models. Built upon the original ModelNet40 dataset, ModelNet40-E injects LiDAR-like, physically-motivated noises, reflecting range-dependent and angle-dependent measurement errors as well as outlier behavior. Each point is annotated with ground-truth uncertainty parameters (µ, σ), enabling a principled study of predictive confidence.

The authors evaluate several point-cloud classifiers under noise conditions and analyze three complementary aspects: (1) robustness (accuracy), (2) calibration (ECE), and (3) uncertainty awareness (AUROC and Pearson correlation with ground-truth σ). Results reveal that models with high clean accuracy might not be the best calibrated or most uncertainty-aware.

**Strengths:**

(+) The authors point out that point-cloud benchmarks typically assess only accuracy and ignore uncertainty—a key limitation for safety-critical robotics and autonomous driving applications. The proposed benchmark directly addresses this issue.

(+) Multiple canonical and modern architectures are trained and tested under unified settings, and the analysis spans calibration curves, AUROC, and correlation metrics, offering some empirical insights.

**Weaknesses:**

(-) While the benchmark is thoughtfully designed, its core contributions (benchmark extension and noise simulation) are largely incremental with respect to prior corruption studies such as ModelNet-C or PointCloud-C. The added uncertainty annotations and LiDAR noise formulation, though practical, do not represent a fundamentally new methodological direction.

(-) Although the manuscript aims at realism, all data remain synthetic (based on CAD models). Without real sensor validation or comparisons to real LiDAR scans, it is difficult to verify whether the simulated noise distribution truly matches physical sensor characteristics.

(-) The manuscript focuses on point-cloud classification only. There is no demonstration of how the benchmark generalizes to segmentation, detection, or other downstream perception tasks, which slightly narrows its community impact.

(-) The manuscript fixes noise parameter ranges (Table 1) without analyzing how these design choices affect calibration or correlation results. Such a discussion would strengthen the benchmark’s credibility.

(-) Lack of qualitative examples. Visualizations of noisy vs. clean samples or examples of miscalibrated predictions would help readers intuitively grasp where uncertainty estimation succeeds or fails.

**Questions:**

Please see the Weaknesses section.

---

### Official Review · Reviewer_CTBr · 2025-11-02

**Soundness:** 2
**Presentation:** 2
**Contribution:** 2
**Rating:** 4
**Confidence:** 4

**Summary:**

The paper introduces ModelNet40-E, a version of ModelNet40 augmented with physically motivated LiDAR-like noise at three severities (Light/Moderate/Heavy) plus per-point noise stats (µ bias, σ std) so models can be assessed on accuracy, calibration, and uncertainty awareness—not accuracy alone . Evaluating representative classifiers, the authors show accuracy drops with noise, calibration generally worsens, and some models with lower accuracy exhibit better calibration/uncertainty awareness, arguing these metrics must be considered together.

**Strengths:**

1. The presentation of this paper is clear and easy to follow.

2. Multi-metric reporting: Accuracy, ECE, AUROC for error detection, and Pearson correlation between σ (true measurement uncertainty) and predicted uncertainty—separately on correct-only vs. all samples to isolate “awareness” from trivial confidence collapse.

3. Training-dynamics analysis: Longer training can improve accuracy but harm calibration and uncertainty awareness, reinforcing why deployment metrics must go beyond accuracy.

**Weaknesses:**

1. Benchmark is tied to ModelNet40 geometry and categories; real LiDAR scenes (e.g., autonomous driving) differ in distribution and occlusion patterns, limiting external validity without cross-dataset evidence. There is no visualization either.

2. The LiDAR model is parametric (linear range noise, cosine angle term, uniform outlier sampling). It’s plausible but may not capture sensor-specific phenomena (e.g., intensity-dependent dropout, multi-return behavior) beyond the chosen parameters in Table 1.

3. Limited ablations on noise settings: Although Light/Moderate/Heavy ranges are given (Table 1), there’s no sensitivity study showing how rankings change with different parameter draws or sensor poses beyond the stated sampling scheme.

**Questions:**

Please provide visualization of the new benchmark to give the reviewers a high-level sense. Please also refer to the weakness section for other questions.

---

### Meta-Review · Area_Chair_rkRT · 2026-01-11

**Summary:**

This paper introduces ModelNet40-E, a benchmark designed to evaluate point cloud classification models not only on accuracy but also on calibration and uncertainty awareness. It augments the ModelNet40 dataset with physically motivated LiDAR-like noise at multiple severity levels and provides per-point uncertainty annotations, arguing that these metrics are crucial for safety-critical applications.

Reviewers raise some key concerns:

Reviewer CTBr raises the following concerns (1) the benchmark is tied to ModelNet40's synthetic geometry and categories, limiting external validity for real-world LiDAR scenes (e.g., autonomous driving) without cross-dataset validation or visualizations. (2) The LiDAR noise model, while plausible, may not capture sensor-specific phenomena (e.g., intensity-dependent dropout). (3) Limited ablation studies on noise parameters; no sensitivity analysis shown for how rankings change with different parameter draws or sensor poses.

Reviewer guUM: (1) Core contributions (benchmark extension, noise simulation) are incremental compared to prior corruption studies (e.g., ModelNet-C). (2) Data remains synthetic without validation against real sensor data, making it difficult to verify noise realism. (3) Focus is limited to classification; no demonstration for segmentation or detection tasks, narrowing community impact. (4) Lacks qualitative examples/visualizations of noisy samples or miscalibrated predictions.

Reviewer QtXM: (1) Benchmark is synthetic and small-scale; conclusions may not transfer to large-scale real-world LiDAR datasets. (2) Limited architectural diversity; evaluation excludes self-supervised or pretraining-heavy backbones (e.g., Point-BERT). (3) Noise realism is simulated but not validated against real sensor logs. (4) The interpretation of the proposed Pearson correlation metric is not straightforward. (5) Computational overhead of storing per-point annotations is not discussed.

Reviewer k8f4: (1) Missing visual demonstration of dataset samples; no figures to illustrate differences from original point clouds. (2) Lack of details on how point clouds are sampled from 3D models (e.g., via ray casting) to simulate realistic LiDAR sweeps. (3) Uncertainty quantification may be insufficient due to limited noise type and magnitude diversity (only three levels).

**Reviewer Concerns:**

The authors did not provide a rebuttal.

**Reviewer Scores:**

The authors did not provide a rebuttal, and the concerns should remain.

---

### Decision · Program_Chairs · 2026-01-26

Reject